# Resonant terahertz detection using graphene plasmons

Denis A. Bandurin[1], Dmitry Svintsov[2], Igor Gayduchenko[2,3], Shuigang G. Xu[1,4], Alessandro Principi[1], Maxim Moskotin[2,3], Ivan Tretyakov[3], Denis Yagodkin[2,3], Sergey Zhukov[2], Takashi Taniguchi[5], Kenji Watanabe [5], Irina V. Grigorieva [1], Marco Polini[1,6], Gregory N. Goltsman[3], Andre K. Geim[1,4] & Georgy Fedorov[2,3]

Plasmons, collective oscillations of electron systems, can efficiently couple light and electric current, and thus can be used to create sub-wavelength photodetectors, radiation mixers, and on-chip spectrometers. Despite considerable effort, it has proven challenging to implement plasmonic devices operating at terahertz frequencies. The material capable to meet this challenge is graphene as it supports long-lived electrically tunable plasmons. Here we demonstrate plasmon-assisted resonant detection of terahertz radiation by antenna-coupled graphene transistors that act as both plasmonic Fabry-Perot cavities and rectifying elements. By varying the plasmon velocity using gate voltage, we tune our detectors between multiple resonant modes and exploit this functionality to measure plasmon wavelength and lifetime in bilayer graphene as well as to probe collective modes in its moiré minibands. Our devices offer a convenient tool for further plasmonic research that is often exceedingly difficult under non-ambient conditions (e.g. cryogenic temperatures) and promise a viable route for various photonic applications.

[1] School of Physics, University of Manchester, Oxford Road, Manchester M13 9PL, UK. [2] Moscow Institute of Physics and Technology (State University), Dolgoprudny, Russian Federation 141700. [3] Physics Department, Moscow State University of Education (MSPU), Moscow, Russian Federation 119435. [4] National Graphene Institute, University of Manchester, Manchester M13 9PL, UK. [5] National Institute for Materials Science, 1-1 Namiki, Tsukuba 305-0044, Japan. [6] Istituto Italiano di Tecnologia, Graphene Labs, Via Morego 30, 16163 Genova, Italy. These authors contributed equally: Denis A. Bandurin, Dmitry Svintsov. Correspondence and requests for materials should be addressed to D.A.B. (email: bandurin.d@gmail.com) or to G.F. (email: fedorov.ge@mipt.ru)

Selective detection and spectroscopy of THz fields is a challenging task in modern optoelectronics offering a wide range of applications: from security and medical inspection to radio astronomy and wireless communications[1,2]. Among the variety of available detection principles[2], one elegant proposal has always stood out and remained intriguing for more than two decades. The idea is to compress incident radiation into highly-confined two-dimensional plasmons propagating in the field effect transistor (FET) channel and to rectify the induced ac potential using the same device[3]. The FET channel, in this case, acts as a tunable plasmonic cavity with a set of resonant frequencies defined by its length and the density of charge carriers. The implementation of such resonant devices has promised on-chip selective sensing, spectroscopy, mixing, and modulation of THz fields below the classical diffraction limit[3]. However, despite decades-long experimental efforts, the excitation of long-lived plasma oscillations in conventional FETs has proven challenging[4–9] and little evidence of resonant THz detection has been found so far[10–14].

Graphene has recently demonstrated great promise for mid- and far-infrared plasmonics[15–22] and attracted a great deal of attention as a platform for plasmonic radiation detectors[19,23]. With lowering the operation frequency down to the THz domain, the resonant excitation of plasmons becomes exceedingly difficult and can only be achieved if the momentum relaxation rate is below the plasmon frequency, which, in turn, requires ultra-high electron mobility. For this reason, in all graphene-based far-field THz detectors reported so far, the plasma waves—if any—were overdamped, and the devices exhibited only a broadband (non-resonant) photoresponse[5–9,24,25]. As a result, numerous applications relying on resonant plasmon excitation (see e.g. refs. [3,26–29]) remain experimentally yet unrealized.

In this work, we demonstrate this long-sought resonant regime using FETs based on high-quality van der Waals heterostructures. In particular, we employ graphene encapsulated between hexagonal boron nitride (hBN) crystals which have been shown to provide the cleanest environment for long-lived graphene plasmons[20,22]. Antenna-mediated coupling of such FETs to free-space radiation results in the emergence of dc photovoltage that peaks when the channel hosts an odd number of plasmon quarter-wavelengths. Exploiting the gate-tunability of plasmon velocity, we switch our detectors between more than 10 resonant modes, and use this functionality to measure plasmon wavelength and lifetime. Thanks to the far-field radiation coupling, our compact devices offer a convenient tool for studies of plasmons in two-dimensional electron systems under non-ambient conditions (e.g. cryogenic environment and high magnetic fields) where other techniques may be arduous. As an example, we apply our approach to probe plasmons in graphene/hBN superlattices and unveil collective modes of charge carriers in moiré minibands.

## Results

**Graphene-based THz detectors.** There are three crucial steps to consider in the design of resonant photodetectors. First, the incoming radiation needs to be efficiently compressed into plasmons propagating in the FET channel. Second, the channel should act as a high-quality plasmonic cavity, where constructive interference of propagating plasma waves leads to the enhancement of the field strength. Third, the high-frequency plasmon field needs to be rectified into a dc photovoltage. To meet these hard-to-satisfy[5–8] requirements, we fabricated proof-of-concept detectors using high-mobility bilayer graphene (BLG) FETs. To this end, we first applied a standard dry transfer technique to encapsulate BLG between two relatively thin ($d \approx 80$ nm) slabs of hBN[30]. The heterostructure had side contacts (Fig. 1a) which

were extended to the millimeter scale and one of them served as a sleeve of the broadband antenna, Fig. 1c and Supplementary Fig. 3a, b (see Methods). Another antenna sleeve was connected to the top gate covering the FET channel (inset of Fig. 1d). In this coupling geometry, the incident radiation induces high-frequency modulation of the gate-to-channel voltage thereby launching plasma waves from the source terminal[3]. The detector was assembled on a THz–transparent Si wafer attached to a Si lens focusing the incident radiation onto the antenna (Fig. 1b).

We studied four BLG FETs, from 3 to 6 μm in length $L$ and from 6 to 10 μm in width $W$, all exhibiting typical field-effect behavior as seen from measurements of the conductance $G$ (Fig. 1d and Supplementary Fig. 3e). In particular, $G$ is minimal at the charge neutrality point and rises with increasing $V_g$. The mobility of our devices at the characteristic carrier density $n = 10^{12}$ cm$^{-2}$ exceeded 10 m$^2$/Vs and remained above 2 m$^2$/Vs at temperatures $T = 10$ K and 300 K, respectively, as determined from the characterization of a multiterminal Hall bar produced under identical protocol reported in Methods (Supplementary Note 1 and Supplementary Fig. 1).

**Broadband operation.** We intentionally start the photoresponse measurements at the low end of the sub-THz domain, where the plasma oscillations are overdamped (see below). This allows us to compare the performance of our detectors with previous reports[5–8,23]. Figure 2a shows an example of the responsivity $R_a = \Delta U/P$, where $\Delta U$ is the emerging source-to-drain photo-voltage and $P$ is the incident radiation power, as a function of the top gate voltage $V_g$ under irradiation with frequency $f = 0.13$ THz in one of our BLG detectors (see Methods). In good agreement with the previous studies, the $R_a(V_g)$ dependence follows the evolution of the FET-factor $F = -\frac{1}{\sigma}\frac{d\sigma}{dV_g}$, shown in the inset of Fig. 2a. In particular, $R_a$ increases in magnitude upon approaching the charge neutrality point (NP) where it flips sign because of the change in charge carrier type.

We have studied the operation of our detectors at different temperatures and found that $R_a$ grows with decreasing $T$ (bottom inset of Fig. 2a) and reaches its maximum $R_a \approx 240$ V/W at $T = 10$ K due to a steeper $F(V_g)$ at this $T$ (top inset of Fig. 2a). At large positive $V_g$, $R_a$ approaches zero at all $T$, whereas at negative $V_g$, a positive offset is observed (orange rectangle in Fig. 2a). This behavior is common for this type of devices and is related to additional rectification by p–n junctions at the boundaries between the p-doped graphene channel and the n-doped contact regions[24,31,32].

The overall broadband responsivity of our BLG detectors is further improved in transistors with stronger nonlinearity, which can be conveniently parametrized by the FET-factor introduced above. To this end, we took advantage of the gate-tunable band structure of BLG and fabricated a dual-gated photodetector. Simultaneous action of the two gates results in a band gap opening and a steep $F(V_g)$ dependence that, in turn, causes a drastic enhancement of $R_a$ (Supplementary Note 2). The latter exceeded 3 kV/W for a weak displacement field $D$ of 0.1 V/nm (Supplementary Fig. 2b). This translates to the noise equivalent power (NEP) of about 0.2 pW/Hz$^{1/2}$, estimated using the Johnson-Nyquist noise spectral density obtained for the same $D$. The observed performance of our detectors makes them competitive not only with other graphene-based THz detectors operating in the broadband regime[23], but also with some commercial superconducting and semiconductor bolometers operating at the same $f$ and $T$ (Supplementary Table 1).

**Resonant operation.** The response of our photodetectors changes drastically as the frequency of incident radiation is

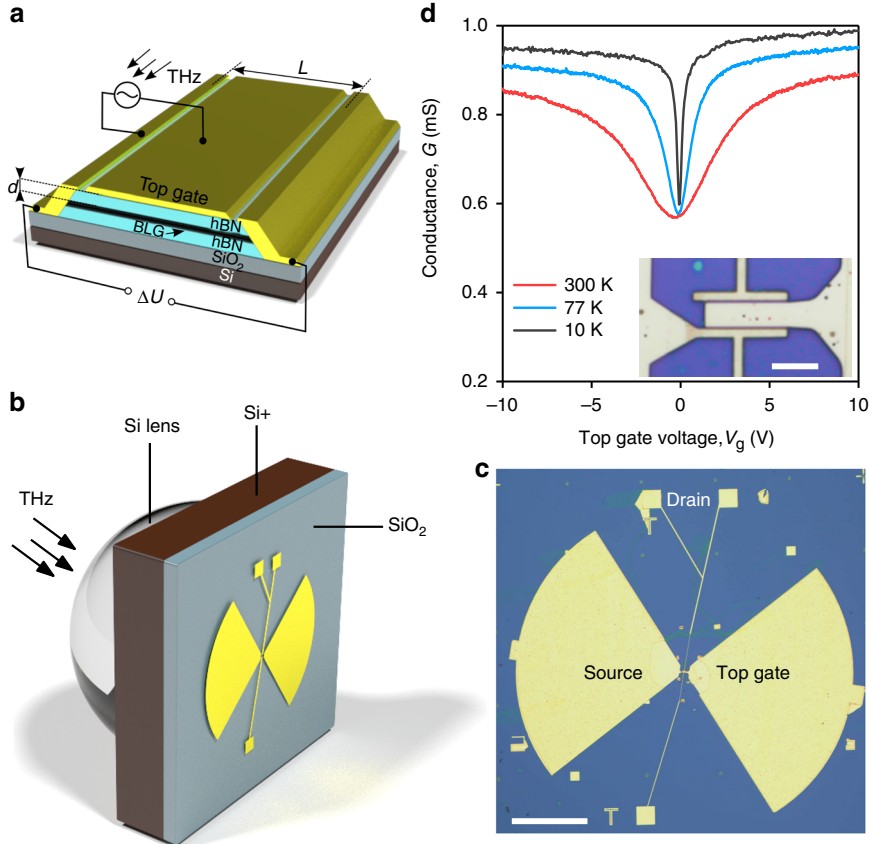

**Fig. 1** Graphene-based THz detectors. **a** Schematics of the encapsulated BLG FET used in this work. **b** 3D rendering of our resonant photodetector. THz radiation is focused to a broadband bow-tie antenna by a hemispherical silicon lens yielding modulation of the gate-to-source voltage, as indicated in **a**. **c** Optical photograph of one of our photodetectors. Scale bar is 200 μm. **d** Conductance of one of our BLG FETs as a function of the gate voltage $V_g$, measured at a few selected temperatures. Inset: zoomed-in photograph of **c** showing a two-terminal FET with gate and source terminals connected to the antenna. Scale bar is 10 μm

increased. Figure 2b shows the gate voltage dependence of $R_a$ recorded in response to 2 THz radiation. In stark contrast to Fig. 2a, $R_a$ exhibits prominent oscillations, despite the fact that $F$ as a function of $V_g$ is featureless (black curve in Fig. 2b). The oscillations are clearly visible for both electron and hole doping and display better contrast on the hole side, likely because of the aforementioned p–n junction rectification. Resonances are well discerned at 10 K, although they persist up to liquid-nitrogen $T$, especially for $V_g < 0$. A further example of resonant operation of another BLG device is shown in Supplementary Note 3.

We have also studied the performance of our detectors at intermediate frequencies and found that the resonant operation of our devices onsets in the middle of the sub-THz domain (Supplementary Note 4). In particular, we have found that at $f = 460$ GHz, the resonances are already well-developed (Supplementary Fig. 4). At such low $f$, only two peaks in the photoresponse (one for electrons and one for holes) are observed for the same gate voltage span as in Fig. 2b along with an apparent increase of their full width at half-height. These observations are in full agreement with the plasmon-assisted photodetection model discussed below.

**Plasmon resonances in graphene FETs**. We argue that the observed peaks in the photoresponse emerge as a result of plasmon resonance in the FET channel. To this end, we model

our FET as a plasmonic Fabry-Perot cavity endowed with a rectifying element. This results in responsivity given by

$$R_a = \frac{R_0}{|1 - r_s r_d e^{2iqL}|^2},\qquad(1)$$

where $R_0$ is a smooth function of carrier density $n$ and frequency $f$ that depends on the microscopic rectification mechanism, $r_s$ and $r_d$ are the wave reflection coefficients from the source and drain terminals, respectively, and $q$ is the complex wave vector governing the wave propagation in the channel (Supplementary Note 5). In gated 2D electron systems, the relation between the frequency $\omega$ and the real part of the wave vector $q'$ is linear, $\omega = sq'$, where the plasmon phase velocity is

$$s = v_F \sqrt{4\alpha_c k_F d} = \sqrt{\frac{e}{m}|V_g|}.\qquad(2)$$

Here $m$ and $e$ are the effective mass of carriers and the elementary charge respectively, $v_F$ and $k_F = \sqrt{\pi n}$ are the Fermi velocity and the Fermi wave vector, $d$ is the distance to the gate, $\alpha_c = e^2/(4\pi\varepsilon_z\varepsilon_0\hbar v_F)$ is the dimensionless coupling constant and $\varepsilon_z$ is the out-of-plane dielectric permittivity[33,34]. We further note that Eq. (2) is valid for monolayer graphene upon replacement of the effective mass $m$ with the cyclotron mass, $m \to \hbar k_F/v_F$ (see Supplementary Note 6). The latter increases with gate-induced carrier density $n$, thereby limiting the tuning range of $s$ for a given voltage span. In contrast, in the case of BLG, $m$ is nearly constant ($\approx 0.036 m_e$) for experimentally

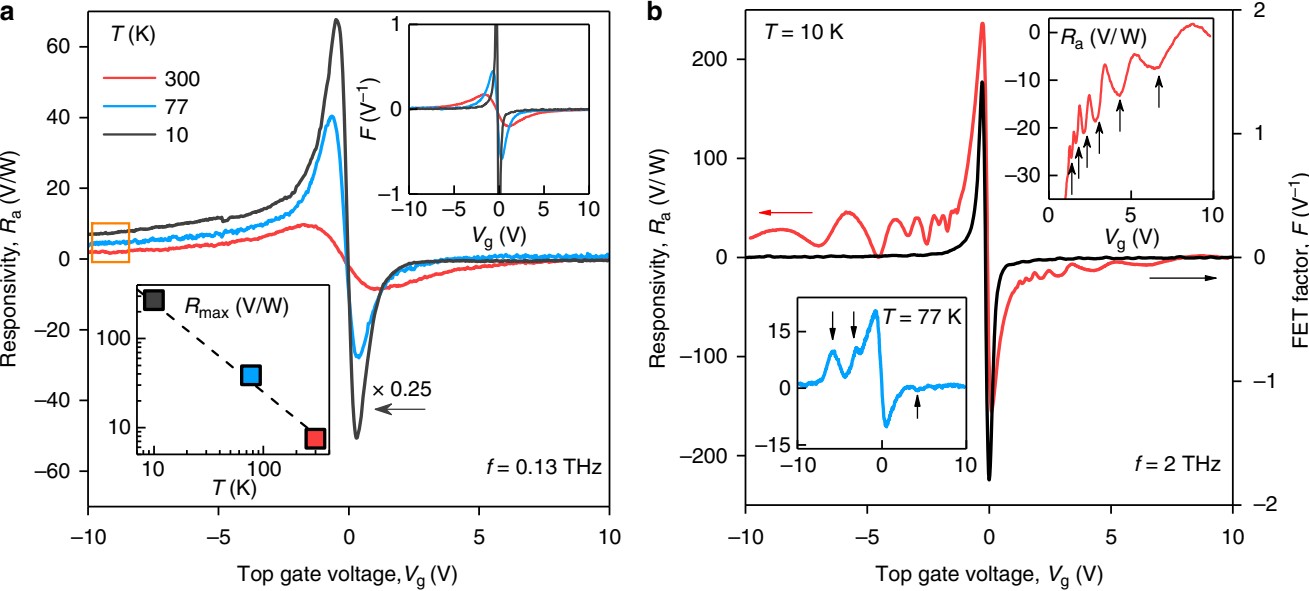

**Fig. 2** Plasmon-assisted THz photodetection. **a** Responsivity measured at $f = 130$ GHz and three representative temperatures. Orange rectangle highlights an offset stemming from the rectification of incident radiation at the p-n junction between the p-doped graphene channel and the n-doped area near the contact. Upper inset: FET-factor $F$ as a function of $V_g$ at the same $T$. Lower inset: maximum $R_a$ as a function of $T$. **b** Gate dependence of responsivity recorded under 2 THz radiation. The upper inset shows a zoomed-in region of the photovoltage for electron doping. Resonances are indicated by black arrows. Lower inset: resonant responsivity at liquid-nitrogen temperature

accessible values of $V_g$, a feature that allows us to vary $s$ over a wider range and thus switch the detector between multiple modes, as we now proceed to show.

It follows from Eq. (1), that the responsivity of our Fabry–Perot rectifier is expected to peak whenever the denominator in Eq. (1) approaches zero. In our devices, the source potential is clamped to antenna voltage, and no ac current flows into the drain, therefore $r_s r_d \approx -1$ (refs. [3,32]). The resonances should therefore occur whenever the real part of the wave number is quantized according to

$$q' = \frac{\pi}{2L}(2k+1), \quad k = 0, 1, 2 \ldots \quad (3)$$

The quantization rule (3) combined with Eq. (2) predicts a linear dependence of the mode number $k$ on $|V_g|^{-1/2}$ which may serve as a benchmark for plasmon resonances in the FET channel. This is indeed the case of our photodetector, as shown in Fig. 3a, e and Supplementary Fig. 3c. The slope of the experimental $k(|V_g|^{-1/2})$ dependence in Fig. 3a matches well the theoretical expectation for a BLG Fabry-Perot cavity of length $L = 6$ μm. At large $|V_g|^{-1/2}$, we find a slight upward trend in the experimental data with respect to the linear dependence. We attribute this trend to deviations of the plasmon dispersion from the linear law at short wavelengths which stem from the non-local relation between electric potential and carrier density[33]. Note that the known non-parabolicity of the BLG spectrum[35] resulting in an increase of $m$ at large density $n$ would bend the dependence in Fig. 3a in the opposite direction.

**Photovoltage-based spectroscopy of 2D plasmons**. The resonant gate-tunable response of our detectors offers a convenient tool to characterize plasmon modes in graphene channels. From Eq. (3) it follows that resonances occur if $L = (2k+1)\lambda_p/4$, where $\lambda_p = 2\pi/q'$ is the plasmon wavelength (Fig. 3b). Using the experimentally observed peak positions, we have determined the

density dependence of $\lambda_p$, shown in Fig. 3c, which flaunts excellent agreement with theory (Supplementary Note 5). The compression ratio $\lambda_p/\lambda_0$ between the plasmon and free-space wavelength ($\lambda_0 = c/f$ and $c$ the speed of light in vacuum) ranges between 1/50 and 1/150, highlighting the ultra-strong confinement of THz fields enabled by graphene plasmons, matching the record value known in the literature[20].

Apart from $\lambda_p$, the resonant responsivity carries information about another valuable characteristic of plasmons, namely, their lifetime, $\tau_p$. The latter is related to the peak width at half-height $\delta$ via (Supplementary Note 7)

$$V_g^{-1/2}/\delta = \omega \tau_p. \quad (4)$$

Using Lorentzian fits to the photoresponse curves (inset of Fig. 3e), we have extracted $\tau_p$ as a function of $n$, shown in Fig. 3d. The lifetime was found to range between ≈0.3 and ≈0.9 ps, which is slightly shorter than the transport time $\tau_{tr} \approx 2$ ps as extracted from the mobility, $\tau_{tr} = m\mu/e$ (Supplementary Note 1). The corresponding quality factor, $Q = 2\pi f \tau_p$, was found to vary between 4 and 11 for $f = 2$ THz, and between 0.2 and 0.7 for $f = 0.13$ THz, see Fig. 3d. The latter implies that it is unreasonable to expect resonant photoresponse of such detectors in the GHz range, and they can only operate in the broadband (non-resonant) regime, in accordance with the data in Fig. 2a. On the contrary, the resonant responsivity should become more profound at higher frequencies of the THz window and can be further enhanced in graphene FETs of higher quality, such as those using graphite gates to screen remote charge impurities[36].

**Miniband plasmons in graphene/hBN superlattices**. The approach demonstrated above is universal and can be applied to studies of plasmons in arbitrary high-mobility 2D systems embedded in FET channels, as we now proceed to show for the case of devices made of BLG/hBN moiré superlattices[37].

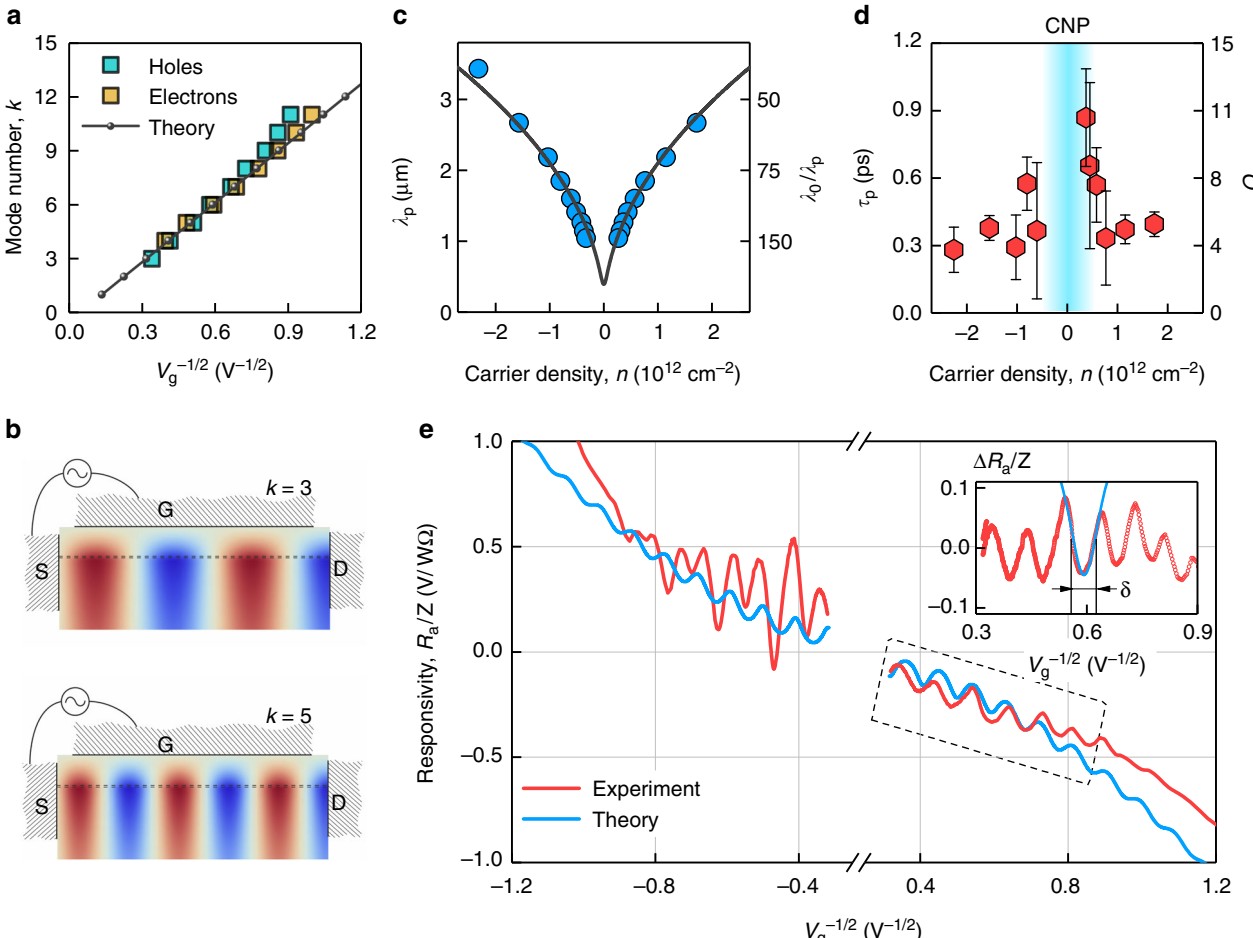

**Fig. 3** Plasmon resonances in encapsulated graphene FET. **a** Mode number $k$ as a function of $V_g^{-1/2}$ (symbols). Solid line: theoretical dependence for $L = 6\,\mu m$, $m = 0.036m_e$, and $f = 2\,THz$. The first mode supported by our Fabry-Pérot plasmonic cavity corresponds to $k_{min} = 3$; the fundamental mode with $k = 0$ is beyond the accessible gate voltages. **b** Examples of high-frequency potential distribution in the plasmon mode (real part) under resonant conditions for given $k$. Brown and blue colors represent positive and negative values of electrical potential, respectively. S, G, and D stand for source, gate, and drain terminals, respectively. **c** Experimental (symbols) and calculated (solid line) plasmon wavelengths $\lambda_p$ as a function of carrier density, as obtained from **a**. The corresponding value of the inverse compression ratio, $\lambda_0/\lambda_p$, for $f = 2\,THz$ is given on the right axis. **d** Plasmon lifetime $\tau_p$ and quality factor $Q$ as obtained from the width of the resonances shown in **e**. Error bars stem from the fitting procedure. **e** Experimental and calculated responsivities as functions of $V_g^{-1/2}$, normalized to the effective antenna impedance $Z = V_a^2/P$ relating the incident power to the resulting gate-to-channel voltage $V_a$. The theoretical Dyakonov-Shur dependence (Supplementary Note 9) was obtained by using characteristic $\tau_p = 0.6\,ps$ from **d**. Inset: normalized responsivity $R_a/Z$ after the subtraction of a smooth non-oscillating background. The solid blue line is the best Lorentzian fit to the data, with $\delta = 0.1\,V^{-1/2}$, which translates to $\tau_p = 0.5\,ps$

Figure 4b and Supplementary Fig. 3c show examples of $R_a$ as a function of $V_g$ recorded in our superlattice devices in response to 2 THz radiation. As in the case of plain BLG, the overall evolution of the superlattice responsivity $R_a$ $(V_g)$ follows that of the FET-factor (black curve) modulated by the plasmon resonances. Note the total number of resonances is smaller due to the shorter FET channel (cf. Supplementary Fig. 3c) and they are visible only for $V_g < 0$, presumably due to a stronger nonlinearity in this detector for negative doping (in another superlattice FET, the resonances were well-observed for both $V_g$ polarities as shown in Supplementary Fig. 3c). Importantly, the FET-factor in these devices is, in turn, a more complex function of $V_g$ (cf. inset of Fig. 2a) due the presence of secondary neutrality points (sNP) stemming from a peculiar band structure of the BLG/hBN superlattice. The latter is characterized by narrow minibands emerging in the vicinity of the $\tilde{K}/\tilde{K}'$-points of the superlattice Brillouin zone[37] (Fig. 4c). The sNPs are clearly visible as peaks in the FET resistance which appear around $V_g = \pm 10\,V$ (Fig. 4a).

A striking feature of the superlattice photoresponse is the resonances appearing when the Fermi level is brought close to the sNP (pink arrows in Fig. 4b). The resonances are of opposite sign with respect to those observed near the main NP (blue arrows), which indicates that they originate from the plasmons supported by the charge carriers of the opposite type (cf. Fig. 2b). Since the latter are hosted by the minibands near the $\tilde{K}/\tilde{K}'$-points of the superlattice Brillouin zone (Fig. 4c), our measurements provide evidence for miniband plasmons that were long identified theoretically[38] but remained elusive in experiment. To date, the experimental studies of superlattice plasmons have been only performed at room temperature using scattering-type scanning near field microscopy operating in the mid-IR domain[39]. The mid-IR excitation energy (10 μm ≈ 120 meV) is high enough to induce interband absorption close to the sNP, which hampers the observation of plasmons in superlattice minibands[39]. In contrast, our approach relies on the low-energy excitations (2 THz ≈ 8 meV), is applicable at cryogenic temperatures, and, therefore,

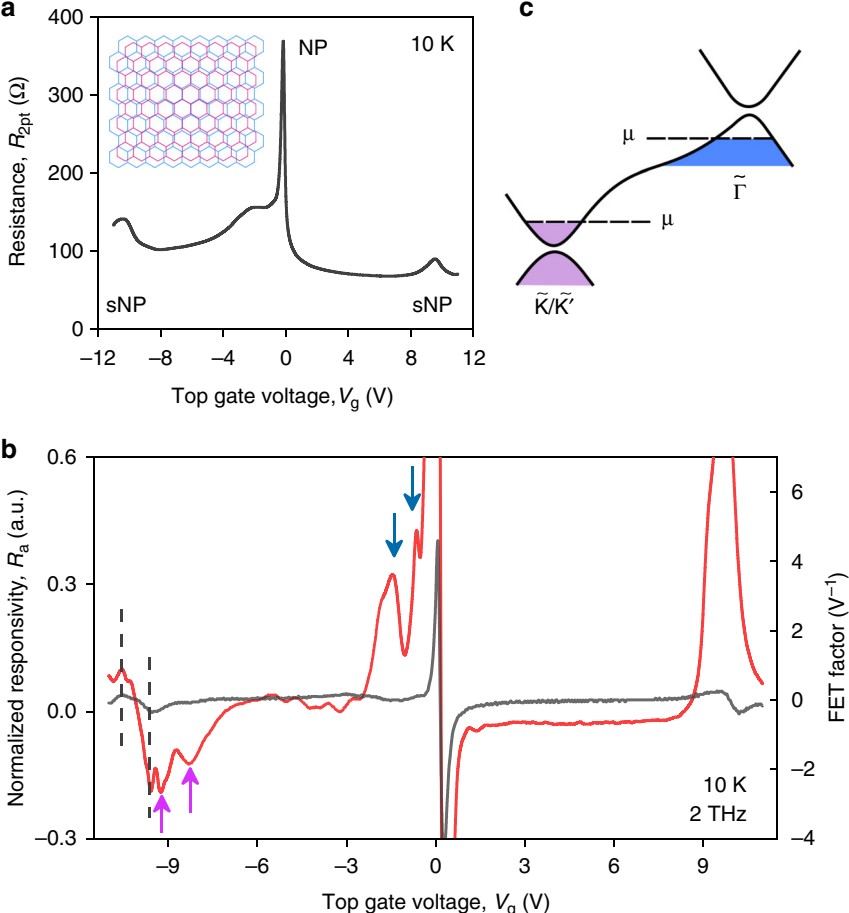

**Fig. 4** Miniband plasmons in BLG/hBN moiré superlattices. **a** Two-terminal resistance of one of our BLG/hBN superlattice devices as a function of $V_g$ measured at given $T$. Inset: illustration of the BLG/hBN superlattice demonstrating a mismatch between graphene and hBN lattice constants. For simplicity, only one graphene layer is shown. **b** Normalized responsivity (red) and the FET-factor (black) as a function of $V_g$ measured in the same device as in **a**. Dashed lines trace $V_g$ where the FET-factor reaches extreme values in the vicinity of the sNP. Pink (blue) arrows point to the resonant peaks near the secondary (main) NP. $L = 3\,\mu m$. **c** Schematic representation of the BLG/hBN superlattice band structure. In the vicinity of the $\tilde{\Gamma}$-point (blue), BLG supports propagation of the ordinary plasma waves. Miniband THz plasmons emerge when the chemical potential approaches the sNP (pink)

paves a convenient way for further studies of miniband plasmonics.

## Discussion

Resonant responsivity is a universal phenomenon in ultra-clean graphene devices and is expected to be independent of the physical mechanisms behind the rectification of the ac field into a dc photovoltage. Nevertheless, it is important to establish possible nonlinearities responsible for the rectification, for example, in order to be able to increase the magnitude of responsivity.

We first note that the aforementioned asymmetry in $R_a (V_g)$ between electron and hole doping indicates rectification at the $p$–$n$ junction formed in vicinity of the contacts. This rectification usually appears due to the thermoelectric effect arising as a result of non-uniform sample heating and the difference between the Seebeck coefficients in the graphene channel and contact regions[21,24,32,40] (Supplementary Note 8). However, $R_a$ remains finite even for $V_g > 0$, where both channel and contact areas are $n$-doped. This indicates that alternative rectification mechanisms are also involved.

Another commonly accepted mechanism is the rectification arising as a result of the simultaneous action of longitudinal high-frequency field and modulation of channel conductivity, also known as resistive self-mixing[2]. The latter can be enhanced by the

dc photovoltage that balances the difference between electron kinetic energies at the source and drain terminals[2,3], similar to Bernoulli's law for classical fluids. Both mechanisms are combined into so-called Dyakonov-Shur (DS) rectification[3] and result in $R_0$ proportional to the sensitivity of the conductivity to the gate voltage variation[4], given by the $F$-factor introduced above (Supplementary Note 9). In Fig. 3e we compare the resonant photoresponse of our BLG photodetector with the responsivity expected from the DS model[3] assuming an average $\tau_p \sim 0.6\,ps$, as found from Fig. 3d, and using the effective antenna impedance $Z$ as the only fitting parameter. The two curves show the same functional behavior and match quantitatively for the n-doped case (where the p–n junction is absent) and $Z \approx 74\,\Omega$, a value close to that expected from the equivalent circuit design[32]. We further note, that although the original DS proposal was based on the hydrodynamic electron transport[41,42], an identical photoresponse is expected outside the hydrodynamic window as it follows from the analysis of graphene's nonlinear conductivity[43].

Last but not least, we note that while the overall trend of the responsivity is well-described by the model introduced above, the values of $\tau_p$ extracted from the peak width at half-height are found to be below the momentum relaxation time. This suggests that other mechanisms of resonance broadening are also involved. In particular, leakage of plasma waves into metal

contacts[44] and electromagnetic dissipation in antenna may also contribute to the apparent resonance width. We have found that respective contributions to $\tau_p^{-1}$ are most pronounced at large carrier densities and small harmonic numbers (Supplementary Figs. 6 and 7), in agreement with experimental data in Fig. 3d. Elimination of these damping channels, e.g. with Schottky/tunnel contacts and low-impedance antennas, may extend the resonant detection down to tens of gigahertz[45]. Other dissipation channels such as electron viscosity[46,47] and interband absorption[18] should be most pronounced at higher-order harmonics and in the vicinity of the NP, as opposed to the data in Fig. 3d, and are unlikely relevant to the present study.

In conclusion, we have shown that high-mobility graphene FETs exploiting far-field coupling to incoming radiation can operate as resonant THz photodetectors. In addition to their potential applications in high-responsivity detection and on-chip spectroscopy of the THz radiation, our devices also represent a convenient tool to study plasmons under conditions where other approaches may be technically challenging. Due to their compact size and far-field coupling, our photodetectors can easily be employed to carry out plasmonic experiments in extreme cryogenic environments and in strong magnetic fields, as well in studies of more complex van der Waals heterostructures. As an example, we have demonstrated the use of our approach to reveal low-energy plasmons hosted by moiré minibands in BLG/hBN superlattices. The method has a strong potential for studies of collective modes in magnetic minibands which have recently gained a great level of attention[48].

## Methods

**Device fabrication**. All our devices were made of BLG. BLG was first encapsulated between relatively thick hBN crystals using the standard dry-peel technique[30]. The thickness of the top hBN was measured by atomic force microscopy. The stack was then deposited either directly on top of a low-conductivity boron-doped silicon wafer capped with a thin oxide layer (500 nm) or on a predefined back gate electrode. The resulting van der Waals heterostructure was patterned using electron beam lithography to define contact regions. Reactive ion etching was then used to selectively remove the areas unprotected by a lithographic mask, resulting in trenches for depositing electrical leads. Metal contacts to graphene were made by evaporating 3 nm of Cr and 60 nm of Au. Afterwards, a second e-beam lithography was used to design the top gate. The graphene channel was finally defined by a third round of e-beam lithography, followed by reactive ion etching etching using Poly(methyl methacrylate) and gold top gate as the etching mask. Finally, we used optical photolithography to pattern large antenna (spiral or bow-tie) sleeves connected to the source and the top-gate terminals, followed by evaporation of 3 nm of Cr and 400 nm of Au. Antennas were designed to operate at an experimentally accessible frequency range.

**Photoresponse measurements**. Photoresponse measurements were performed in a variable temperature optical cryostat equipped with a polyethylene window that allowed us to couple the photodetector to incident THz radiation. The latter was focused to the device antenna by a silicon hemispherical lens attached to the silicon side of the chip (Fig. 1b). The transparency of the chips to THz radiation over the entire temperature and frequency range was verified in transmission experiments using a home-made optical cryostat coupled to the THz spectrometer. Photovoltage measurements were performed using either a standard lockin amplifier synchronized with a chopper rotating at 1 kHz frequency, positioned between the radiation source and the cryostat window, or by a home-made measurement board.

In order to study the photoresponse of our detectors at different frequencies, we used three radiation sources. Sub-THz radiation was provided by two backward wave oscillators (BWO) generating $f = 0.13$ THz and $f = 0.46$ THz. For higher frequencies, a quantum cascade continues wave laser based on a GaAs/Al$_{0.1}$Ga$_{0.9}$As heterostructure emitting $f = 2.026$ THz radiation was used.

The responsivity of our devices was calculated assuming that the full power delivered to the device antenna funnelled into the FET channel. The as-determined value provides the lower bound for our detectors' responsivity and is usually referred to as extrinsic. The calculation procedure consisted of a few steps. First the source-to-drain voltage $U_{dark}$ was measured as a function of $V_g$ in the dark. Then, the dependence of the source-to-drain voltage $U_{SD}$ on $V_g$ was recorded under illumination with THz radiation. The difference $\Delta U = U_{SD} - U_{dark}$ formed the photovoltage. At the next stage, we measured the full power $P_{full}$ delivered to the cryostat window using Golay cell. The responsivity was then calculated as $R_a = \Delta U/P$, where $P \approx P_{full}/3.5$ is the power delivered to the device antenna after accounting for losses in the silicon lens and the cryostat optical window ($\approx 5.5$ dB). All the measurements reported above were performed in the linear-in-$P$ regime.

The performance of our detectors outside the linear regime is discussed in Supplementary Note 10 and reported in Supplementary Fig. 8.

## Data availability
The data that support the findings of this study are available from the corresponding author upon reasonable request.

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

## Acknowledgments

Device fabrication and Manchester's part of the work was supported by the European Research Council, the Graphene Flagship and Lloyd's Register Foundation. The work at the MSPU (Photoresponse measurements) has been carried out with the support of the Russian Science Foundation (project No. 17-72-30036). D.A.B. acknowledges financial support from Leverhulme Trust. Experimental work of M.M. (transport measurements) was supported by Russian Science Foundation (Grant 18-72-00234). M.P. was supported by the European Union's Horizon 2020 research and innovation program under grant agreement No. 785219 - GrapheneCore2. Modelling of antenna electrodynamics was supported by RFBR (Project 18-29-20116). Theoretical work of D.S. was supported by the grant 16-19-10557 of the Russian Science Foundation. Photoresponse measurements have been performed using quantum cascade laser fabricated by A. Valavanis in the group of Prof. Dragan Indjin at the University of Leeds (UK). We thank A. Tomadin, R. Krishna Kumar, A. Berdyugin, L. Levitov, and V. Fal'ko for fruitful discussions.

## Author contributions

D.A.B. and G.F. designed and supervised the project. S.G.X. and I.G. fabricated the devices. Photoresponse measurements were carried out by I.G., M.M., and D.A.B. Data analysis was performed by D.A.B. and D.S. Theory analysis was done by D.S. The manuscript was written by D.A.B. and D.S. with input from I.V.G., M.P., A.P., and A.K.G. Experimental support was provided by I.T., D.Y., S.Z., and G.G.; T.T. and K.W. grew the hBN crystals. All authors contributed to discussions.

## Additional information

**Competing interests:** The authors declare no competing interests.

