## [Peer Review File · Nature Communications]

Reviewers' comments:

Reviewer #1 (Remarks to the Author):

This paper experimentally demonstrates the plasmon-resonant terahertz rectification/detection in a gated graphene-channel field effect transistor structure. A high quality of graphene, which is encapsulated by h-BN layers, acts as the two-dimensional (2D) carrier transit channel as well as the 2D plasmon Fabry-Perot cavity between source and drain edge-electrode terminals. The gate bias voltage tunes the graphene carrier density and the plasmon velocity. Thus it tunes the plasmon harmonic modes of the resonant frequencies. The fabricated device exhibits temperature-dependent fine ambipolar current-voltage (IV) characteristics, demonstrating higher carrier mobility and longer momentum relaxation time (τ) at lower temperatures. When an electromagnetic wave is irradiated, the graphene plasmons are excited via an integrated log-spiral antenna. Depending on the $\omega\tau$ values non-resonant (at 0.34 THz when $\omega\tau < 1$) or resonant (at 2 THz when $\omega\tau > 1$) photoresponse was observed at cryogenic temperatures (at 10K and up to 77K). The photoresponse under plasmon-resonant detection conditions to the 2-THz radiation exhibits clear multiple peaks corresponding to the harmonic modes of plasmon resonant frequencies. As far as the reviewer's understanding this is the first result of resonant terahertz detection obtained using graphene 2D plasmons. However, similar results using other semiconductor 2D plasmons in FETs have already been experimentally demonstrated by using InGaAs/InAlAs quantum well (QW) high-electron mobility transistors (HEMTs), first in Ref. [1] in 2008 and then followed by Ref. [2] in 2013. The authors never mention/cite these prior works, but claim this is the first experimental demonstration of plasmon-resonant terahertz detection, which is WRONG ! In particular, the results in Ref. [1] was similar observation of multiple higher harmonic mode peaks to the 0.54-THz radiation at 10 K and up to ~35K with quality factors of 5 to 9. In Ref. [2] photoresponse showed only the fundamental resonant mode but at rather higher temperatures up to 125K to the 0.29-THz radiation that was not able to measure in the resonant mode but in the non-resonant mode in this work. Be reminded that the resonant detection is obtained when the cavity quality factor is larger than 1, which is simply given by $\omega\tau > 1$. 2D plasmons in an InGaAs channel, whose carrier momentum relaxation time τ is shorter than that in graphene, can make the resonant detection to the 0.29-THz radiation at even higher temperatures. The authors claim the superior carrier transport property of graphene but could not obtain the resonant detection to the 0.34-THz radiation. This mismatch cannot be understood by only idealistic factors with a simple modeling.

Even if it is revised so as to mention all those prior works and discuss the results quantitatively in comparison with them in InGaAs/InAlAs QWs, no clear superiority of this work using graphene cannot be found.

Judging from the aforementioned facts and reasons the reviewer concludes that the paper does not have any merits for publication.

[1] S. Boubanga-Tombet et al., Appl. Phys. Lett. 92, 212101 (2008).

doi: 10.1063/1.2936077

[2] T. Otsuji et al., IEEE Trans. Terhz. Sci. Technol. 3, 63-71 (2013).

doi: 10.1109/TTHZ.2012.2235911

Reviewer #2 (Remarks to the Author):

The authors report studies of bilayer graphene detectors in FET antenna-coupled configuration. Similar studies have been reported before by the W. Knap group and collaborators (refs. 4-19). Reference 6 reports studies in bilayer graphene, similar to the authors, but at a frequency of 400 GHz. The authors recognise these prior works. On a similar structure, but on reportedly higher

quality graphene layers, they perform comparative studies as functions of the temperature (previous reports concern exclusively room temperature investigations) and the frequency of the incoming radiation, from 100 GHz to 2 THz. At low temperature and at high frequency (2 THz) the authors report the excitation of standing plasmon waves in the FET channel, which have never been observed before in such systems. By modelling the corresponding plasma oscillations as a function of the FET gate bias the authors extract the resonant wavelength and damping constant of the plasmon waves. They conclude that because of the typical relaxation time (0.5 ps) the previous work by the Knap group was dealing mainly with overdamped plasma waves, and the Dyakonov and Schur detection mechanism (ref. 3) has never actually been observed until now. This is a strong message to the community.

The reported experimental studies are clear and the observation of plasma standing waves is convincing: i.e. the recovery of the expected plasma dispersion as a function of the gate voltage (Fig. 3a). In the supporting document, the authors provide a sound model for the plasma Fabry-Perot effect in the responsivity as a function of V_g , and they discuss extensively the rectifying mechanisms in their device. I believe that this manuscript is of sufficient significance so that the publication in Nature Communications is justified, after the following issues have been addressed:

#1. While It is true that the electric field of the plasma waves is intrinsically strongly “compressed” with respect to the free space wavelength (1/150 in the present case), the conversion between the radiation field and the plasmon waves in the FET channel is mediated mainly by the antenna element. Therefore, the “compression” of the incoming radiation into the FET channel is a trivial function that has been realized in many THz devices. My feeling is that this trivial effect is highly overstated in the introductory paragraphs.

#2 It is not clear to me what is the coupling efficiency of the antenna, in terms ratio between the incoming power and the dropped power in the FET channel. Are the values of the responsivity provided with respect to the measured output power measured from the sources, or they represent the internal responsivity evaluated from the model in the supplementary material? More information is needed for the responsivity calibration in the Methods or the Supplementary material section.

#3 While clearly visible, the plasma oscillations in the responsivity curves (i.e. 2b, 3e) have very small contrast, on the order of few percents only. I do not see how such small features can be referred to as “resonant” or “selective” detection: indeed, the overall behaviour of the responsivity versus V_g is dominated by the envelope factor R_0 in Eq.(1); the maximum values of the responsivity are always found at zero gate bias, away from the plasma oscillations which appear as secondary features. The term “resonant detection” in the title is thus extremely misleading and should be modified: instead of “resonant terahertz detection” probably another formulation, such as “Evidence of plasma resonances in bilayer graphene FET” or similar should be used.

#4 Seen as a detector this device do not seem competitive with other standard commercial THz detectors, such as germanium bolometers, which have typical NEP $< 1 \text{ pW/Hz}^{0.5}$ at 4.5 K. With that respect, as well as according to the comment #2 the statement for “high-responsivity selective THz detection” in the conclusion part should be removed or strongly moderated.

Reviewer #3 (Remarks to the Author):

Review of Nature Communications MS “Resonant Terahertz Detection Using Graphene Plasmons” by Bandurin et al.

This work presents an interesting study of resonant and non-resonant THz detection based on plasmons in bilayer graphene (BLG) – hBN based FET structure. The high quality BLG with mobility

up to $10 \text{ m}^2/\text{Vs}$ (at $T = 10 \text{ K}$) at the doping density of $n = 10^{12} \text{ cm}^{-2}$ was used for the manufactured devices. At sub-THz frequencies (130 GHz), corresponding to the field oscillation periods longer than the plasmon damping time, the detection showed a well-known non-resonant overdamped behavior corresponding to the Q factor less than 1 in the FET channel. However, as the incident field frequency was increased to 2 THz, the clear resonant behavior of the detector was demonstrated, with the reasonably high Q factor between 4 and 11, ensuring significant resonant plasmon confinement in the FET channel.

I find this work interesting and timely. As the authors write, such an arrangement indeed allows the study of plasmon physics in confined geometries and under non-ambient conditions (e.g. low temperatures and high B-fields), without the need of tip-based spectroscopies.

The paper is well-organized and is easy to read.

I however question the application motivation for such devices, which only show the novel resonant behaviour at cryogenic temperatures.

Further, when the authors mention the state of the art, they avoid the direct comparison between their devices and the state of the art in the literature. I strongly recommend to update the manuscript with the table comparing such key parameters of the plasmonic detectors as NEP and the plasmon confinement λ_0 / λ_D (for the corresponding operation temperature) for their devices and for the literature state of the art.

Additional comments:

The authors write that in monolayer graphene the electron mass is dependent on the electron density. This is rather confusing, since in the monolayer graphene, within the Dirac cone (i.e. in the range of about $\pm 1.5 \text{ eV}$ with respect to the neutrality point) the electron mass is zero and the band velocity is constant. I guess this statement should be revised or clarified.

Further, what peak temperatures do the electron reach, and what is the ratio between the peak temperature and the Fermi temperature, in the presented THz detectors? It is well known that the transient electron heating in the THz fields can strongly modulate the conductivity in graphene (see e.g. Nature 561, 507 (2018), Nature Commun. 6, 7655 (2015)). Intuitively, this would positively add to the resistive self-mixing contribution in the FET, further enhancing the detector efficiency. Quite a strong plasmonic confinement in the channel might indeed lead to a significant electron heating even at moderate powers of the incident THz signal. Has this effect been considered? Is it of relevance, or the relative temperature increase is small and can be neglected? A comment on this would be helpful.

Typos:

- 1) Photovoltage-based spectroscopy of 2D plasmons
- 2) ... proportional to the the sensitivity
- 3) Reference list should be checked for accuracy (spelling of authors' names etc)

Reviewer #1:

This paper experimentally demonstrates the plasmon-resonant terahertz rectification/detection in a gated graphene-channel field effect transistor structure. A high quality of graphene, which is encapsulated by h-BN layers, acts as the two-dimensional (2D) carrier transit channel as well as the 2D plasmon Fabry-Perot cavity between source and drain edge-electrode terminals. The gate bias voltage tunes the graphene carrier density and the plasmon velocity. Thus it tunes the plasmon harmonic modes of the resonant frequencies. The fabricated device exhibits temperature-dependent fine ambipolar current-voltage (IV) characteristics, demonstrating higher carrier mobility and longer momentum relaxation time (τ) at lower temperatures.

When an electromagnetic wave is irradiated, the graphene plasmons are excited via an integrated log-spiral antenna. Depending on the $\omega\tau$ values non-resonant (at 0.34 THz when $\omega\tau < 1$) or resonant (at 2 THz when $\omega\tau > 1$) photoresponse was observed at cryogenic temperatures (at 10K and up to 77K). The photoresponse under plasmon-resonant detection conditions to the 2-THz radiation exhibits clear multiple peaks corresponding to the harmonic modes of plasmon resonant frequencies.

We thank the reviewer for careful reading of our manuscript.

As far as the reviewer's understanding this is the first result of resonant terahertz detection obtained using graphene 2D plasmons. However, similar results using other semiconductor 2D plasmons in FETs have already been experimentally demonstrated by using InGaAs/InAlAs quantum well (QW) high-electron mobility transistors (HEMTs), first in Ref. [1] in 2008 and then followed by Ref. [2] in 2013.

The attempts to demonstrate resonant photoresponse in InGaAs/InAlAs quantum well field effect transistors have been indeed performed in the works mentioned by the Reviewer as well as in other earlier papers which we cited in the first paragraph of our manuscript. Instead of mentioning 2008 and 2013 works pointed by the Reviewer, we have chosen to refer to other results which date back to 2002 (see below).

Nevertheless, let us still comment on the two papers mentioned by the Reviewer. Neither of them provided a quantitative proof that the bulges, observed in the photoresponse, emerge as a result of standing plasma waves in the FET channel. In our work, we have carefully examined the resonances as a function of carrier density and provided an unambiguous evidence of the plasmonic nature of the observed peaks. This follows directly from the flaunting quantitative agreement between the experimental peak positions and those predicted by theory (Fig. 3a). To the best of our knowledge, this has never been reported until now.

The authors never mention/cite these prior works, but claim this is the first experimental demonstration of plasmon-resonant terahertz detection, which is WRONG !

We strongly disagree with the Reviewer's comment as in the introductory section, we cite numerous works describing the use of III-V field effect transistors (FET) for terahertz detection and acknowledge the fact that there have been several attempts to achieve their resonant operation (refs. 4, 10 – 14). Obviously, we cannot cite all the papers related to III-V photodetectors and therefore mentioned only the earliest works (10-11), those which, in our opinion, provided the most reliable evidence of standing plasma waves in III-V FETs (ref.

12-14) and a review covering the topic (4) which, by the way, discusses the work Appl. Phys. Lett. 92, 212101 (2008) in detail.

Next, we would like to refer to our introductory section where we state: "...little evidence of resonant THz detection has been found so far¹⁰⁻¹⁴" which by itself should be enough to address the Reviewer's critics related to our 'claims'.

Last but not least, we are confident that our work is indeed the first demonstration of this long-sought by plasmonic community resonant regime of THz detection in graphene FETs. We remind that most of the previous works dealt with graphene (refs. 5-9) deposited on SiO₂/Si substrates which hampered resonant detection because of the strong plasmon damping. In our work, we provided a detailed recipe to circumvent these limitations and proposed to use such detectors as a tool for fundamental plasmonic studies. As an example, in the revised manuscript we report novel collective modes unveiled in moiré minibands revealed applying our approach to FETs made of graphene/hBN superlattices. These collective modes were long identified theoretically but remained elusive in experiments because of the technological limitations of the existing methods.

In particular, the results in Ref. [1] was similar observation of multiple higher harmonic mode peaks to the 0.54-THz radiation at 10 K and up to ~35K with quality factors of 5 to 9. In Ref. [2] photoresponse showed only the fundamental resonant mode but at rather higher temperatures up to 125K to the 0.29-THz radiation that was not able to measure in the resonant mode but in the non-resonant mode in this work.

We thank the reviewer for providing us with the overview on InGaAs photodetectors.

Be reminded that the resonant detection is obtained when the cavity quality factor is larger than 1, which is simply given by $\omega\tau > 1$. 2D plasmons in an InGaAs channel, whose carrier momentum relaxation time τ is shorter than that in graphene, can make the resonant detection to the 0.29-THz radiation at even higher temperatures. The authors claim the superior carrier transport property of graphene but could not obtain the resonant detection to the 0.34-THz radiation. This mismatch cannot be understood by only idealistic factors with a simple modeling.

We strongly disagree with the Reviewer. We didn't state that we could not obtain the resonant detection at frequencies between those reported in the main text (0.13 and 2 THz) – we simply did not perform such measurements. Nevertheless, by the Reviewer's request, we have carried out such experiments yet at a slightly higher frequency than that requested by the Reviewer, namely at 0.46 GHz (0.34 THz source is unavailable for us) and found that our resonant detectors perform well even at such low f . Away from the charge neutrality point, the responsivity goes through two maxima (stars in Fig. R1) that reflects plasmon resonances in the FET channel as it follows from the comparison with theory (Inset of Fig. R1). In the revised Supplementary Information, we report the responsivity acquired at this intermediate frequency and point that resonant operation of our detectors onsets in the sub-THz domain.

We also emphasize, that in the main text, the low-end of the sub-THz domain was chosen intentionally to characterize the response of our detectors in the overdamped regime and compare its performance with other graphene-based detectors.

Fig. R1. Resonant photoresponse in the sub-THz domain. Normalized to unity responsivity as a function gate voltage measured in one of our BLG detectors at given T and f . Black line represents the FET factor acquired at the same T . Inset: Theory.

Even if it is revised so as to mention all those prior works and discuss the results quantitatively in comparison with them in InGaAs/InAlAs QWs, no clear superiority of this work using graphene cannot be found.

We completely disagree with the Reviewer and below we would like to reiterate the main results of our work. Note, points 4-5 appear in the revised draft as the respective measurements have been performed after the initial submission.

1. We have provided the first evidence of resonant THz detection in graphene FETs.
2. The resonant operation was found to be due to plasmon standing waves confined into the FET channel as we unambiguously prove analysing the gate voltage dependence of more than 10 resonant modes.
3. We have carried out a spectroscopic study of BLG THz plasmon and measure their lifetime. We have further proposed to use photodetectors built using our recipe as a tool for further plasmonic research in cryogenic environment and under strong magnetic fields.
4. As an example, we have studied the response of THz detectors made of BLG/hBN superlattices and unveiled a new type of collective modes existing in their moiré minibands. These collective modes have been known theoretically yet remained elusive in experiments.
5. Last but not least, we have demonstrated that BLG detectors, endowed with an additional back gate have stronger transistor nonlinearity thanks to the gate-tuneable BLG band structure, that resulted in a drastic increase of the detectors' performance. The latter became comparable to commercial state-of-the-art THz bolometers operating at the same f and T as we demonstrate in the revised Supplementary Section 2.

Judging from the aforementioned facts and reasons the reviewer concludes that the paper does not have any merits for publication.

Judging from the aforementioned results 1-5, we believe that our paper fully meets the criteria set by Nature Communications and will be of interest for a broad readership.

Reviewer #2:

The authors report studies of bilayer graphene detectors in FET antenna-coupled configuration. Similar studies have been reported before by the W. Knap group and collaborators (refs. 4-19). Reference 6 reports studies in bilayer graphene, similar to the authors, but at a frequency of 400 GHz. The authors recognise these prior works. On a similar structure, but on reportedly higher quality graphene layers, they perform comparative studies as functions of the temperature (previous reports concern exclusively room temperature investigations) and the frequency of the incoming radiation, from 100 GHz to 2 THz. At low temperature and at high frequency (2 THz) the authors report the excitation of standing plasmon waves in the FET channel, which have never been observed before in such systems. By modelling the corresponding plasma oscillations as a function of the FET gate bias the authors extract the resonant wavelength and damping constant of the plasmon waves. They conclude that because of the typical relaxation time (0.5 ps) the previous work by the Knap group was dealing mainly with overdamped plasma waves, and the Dyakonov and Schur detection mechanism (ref. 3) has never actually been observed until now. This is a strong message to the community.

We thank the carefully reading our manuscript, recognising its importance and this kind assessment.

The reported experimental studies are clear and the observation of plasma standing waves is convincing: i.e. the recovery of the expected plasma dispersion as a function of the gate voltage (Fig. 3a). In the supporting document, the authors provide a sound model for the plasma Fabry-Perot effect in the responsivity as a function of V_g , and they discuss extensively the rectifying mechanisms in their device. I believe that this manuscript is of sufficient significance so that the publication in Nature Communications is justified, after the following issues have been addressed:

We thank the reviewer for this kind assessment and supporting our manuscript to be published in Nature Communications.

#1. While It is true that the electric field of the plasma waves is intrinsically strongly “compressed” with respect to the free space wavelength ($1/150$ in the present case), the conversion between the radiation field and the plasmon waves in the FET channel is mediated mainly by the antenna element. Therefore, the “compression” of the incoming radiation into the FET channel is a trivial function that has been realized in many THz devices. My feeling is that this trivial effect is highly overstated in the introductory paragraphs.

Indeed, the radiation funnelling into the FET channel is ensured solely by the antenna element and this is apparent for researchers working the field of radiation detectors. However, for plasmonics community, especially for those working with near-field techniques (such as SNOM), it has been always challenging to compress light well below the diffraction limit (see refs. 20-22, where the ratio λ_p / λ_0 has become a standard figure of merit for graphene plasmons). By mentioning large compression ratios, we send a message to the plasmonics community, that FET is a convenient tool (no worse than SNOM) for fundamental studies of plasmons as we now also support by revealing a novel type of collective modes in BLG/hBN moiré minibands (see revised Section “Miniband plasmons in graphene/hBN superlattices”). Therefore, we prefer to retain this discussion to attract attention of a broader readership to our manuscript.

In addition, there are several phenomena where the “compression ratio” λ_p / λ_0 becomes indeed an important figure of merit. In particular, it is proportional to the ratio of zero-point electromagnetic fluctuations in the plasmonic resonator. Therefore, it governs the enhancement of radiative decay processes (of atoms, molecules, quantum dots, etc) due to plasmons and therefore needs to be accounted in related studies (e.g. see Koppens, F. H., Chang, D. E., & Garcia de Abajo, F. J. *Nano letters*, 11(8), 3370-3377 (2011)).

#2 It is not clear to me what is the coupling efficiency of the antenna, in terms ratio between the incoming power and the dropped power in the FET channel. Are the values of the responsivity provided with respect to the measured output power measured from the sources, or they represent the internal responsivity evaluated from the model in the supplementary material?

The responsivity which we provide in the main text is referred to as extrinsic as its calculation accounts for the full power measured from the radiation source with some adjustment to account for losses in silicon lens and optical cryostat window as has been discussed in Methods. Note, we refrain from ‘earning’ extra responsivity by normalizing the extrinsic one to the coupling efficiency.

More information is needed for the responsivity calibration in the Methods or the Supplementary material section.

We thank the reviewer for pointing that the responsivity calculation procedure reported in the “Methods” section is rather vague. To make it more transparent, in Methods we provide a more detailed protocol which we followed to determine the responsivity of our detectors.

For convenience, let us reiterate this protocol here. First of all, we examine our devices in the dark. To this end, the source-to-drain voltage U_{dark} is recorded as a function of gate voltage V_g . At the next step, the source-to-drain voltage U_{SD} is measured under continuous illumination with THz radiation. The difference between the two $\Delta U = U_{\text{SD}} - U_{\text{dark}}$ is further referred to as photovoltage. The responsivity R_a is then determined as the photovoltage normalized to the power P delivered to the device antenna which is obtained by measuring the source output power P_{source} and accounting for 5.5 dB losses in silicon lens and optical cryostat window ($P \approx P_{\text{source}}/3.5$). Importantly, for each sample we measure the photovoltage at several values of P_{source} and limit ourselves to the range of power where ΔU scales linearly with P_{source} .

We note, that there is yet another procedure that is also often used to estimate the extrinsic responsivity of THz detectors based on antenna-coupled FETs exposed to the free space radiation. In this case, the responsivity is calculated as $R_a^* = \Delta U S_t / S_a P$, where S_t is the radiation beam spot area and S_a is the detector active area. The latter is usually assumed to be of the order of $\lambda_0^2/4$ while $S_t = \pi r^2$ where r is the radiation spot size radius. Considering that the coupling of our device to the free space radiation is mediated by a hemispherical silicon lens with $r \sim \lambda_0$, two methods yield similar values of R_a .

#3 While clearly visible, the plasma oscillations in the responsivity curves (i.e. 2b, 3e) have very small contrast, on the order of few percents only.

We agree with the Reviewer, that in the as-measured data the resonances have very small contrast because of the steep envelope modulation. However, after such a smooth background is removed (e.g. by dividing the response measured at 2 THz by the non-resonant one recorded at 0.13 THz), the remaining resonant contribution is characterized by a significant contrast which is usually parameterized by the visibility function $(R_{\max} - R_{\min}) / (R_{\max} + R_{\min})$, where R_{\max} and R_{\min} are the maximum and minimum of the responsivity respectively. For some modes, such as that highlighted by the red trace in Fig. R2, the visibility may be as high as 70-100 %. The latter can be further enhanced in the devices with properly adjusted antenna parameters as we now discuss in Supplementary Sections 6.

Fig. R2. Resonant contribution to responsivity. Responsivity as a function of $V_g^{-1/2}$ for one of our BLG devices after a smooth non-oscillating background is removed. $f=2$ THz.

I do not see how such small features can be referred to as “resonant” or “selective” detection: indeed, the overall behaviour of the responsivity versus V_g is dominated by the envelope factor R_0 in Eq.(1); the maximum values of the responsivity are always found at zero gate bias, away from the plasma oscillations which appear as secondary features.

We agree with the Reviewer that the use of word “selective” as an adjective characterizing the observed photoresponse is inaccurate and we have removed it from the concluding section. We however would like to keep the word “resonant” as it is unambiguously related to the plasmon standing waves in the FET channel where the latter acts as a gate-tuneable Fabry-Perot resonator. To the best of our knowledge, there is a solid consensus in the literature on the term used to describe such photoresponse.

We further note, that indeed the envelope function sets the responsivity evolution with the gate voltage and non-surprisingly, the maximum responsivity is found near the neutrality point where the non-linearity is at its extremum (please refer to the FET factor shown in the inset of Fig. 2a). Nonetheless, as we show in Fig. R2, the resonances are well pronounced after such a background is removed. More importantly, the set of peaks is unique for different frequencies of incoming radiation and therefore can be envisioned as a useful feature to construct on-chip spectrometers of THz radiation.

The term “resonant detection” in the title is thus extremely misleading and should be modified: instead of “resonant terahertz detection” probably another formulation, such as “Evidence of plasma resonances in bilayer graphene FET” or similar should be used.

We disagree with the Reviewer. The term resonant detection is well established in the literature on THz detectors and is referred to the resonance of plasmons in the FET channel

with incoming THz radiation. In addition, plasmon quality factors exceeding unity (4-10 in our case) further justify the use of term “resonant” in this context. Thus, since the response of our detectors is strongly affected and modulated by such Fabry-Perot modes we tend to believe that the present title well describes the content of our manuscript and retain the initial title.

#4 Seen as a detector this device do not seem competitive with other standard commercial THz detectors, such as germanium bolometers, which have typical NEP $<1\text{pW}/\text{Hz}^{0.5}$ at 4.5 K. With that respect, as well as according to the comment #2 the statement for “high-responsivity selective THz detection” in the conclusion part should be removed or strongly moderated.

We understand and appreciate this point. The primary goal of our study was to build a proof-of-concept terahertz detector using high-mobility van der Waals heterostructures and demonstrate its resonant operation as it has been a long-sought task in graphene THz plasmonics since the first demonstration of broadband photodetectors based on graphene in 2012 (see Nat. Nanotechnol, 9, 780–793 (2014) for review). In addition, we aimed to propose a simple tool to perform plasmonic studies in graphene-based devices.

Nevertheless, after the initial submission of our manuscript, we have managed to improve the performance of our detectors drastically. In the revised Supplementary Section 2, we include the results of our recent measurements of a similar BLG detector but equipped with an additional back gate terminal. Thanks to the gate-tuneable BLG band structure, the use of a dual-gated architecture allowed us to induce a strong nonlinearity in the FET channel which resulted in broadband responsivity exceeding 3 kV/W and the NEP lowered down to $0.2\text{ pW}/\text{Hz}^{0.5}$ (see Fig. R3) for certain gate voltages. This makes our devices comparable to semiconductor and superconductor hot electron bolometers operating at the same T and f (see Table 1 of the Supplementary Section 2). As per resonant operation, as we argued above, despite slightly lower responsivity, the peaks position is unique for each frequency and therefore can be employed for on-chip THz spectroscopy. We believe that these observations are sufficient to argue the exceptional properties of our devices.

Fig. R3. High-responsivity THz detection by dual-gated BLG field effect transistors. a, Two-terminal resistance as a function of V_{tg} measured in a dual-gated BLG FET for different V_{bg} . Top inset: Schematic of a dual-gated THz detector. Bottom inset: Optical photographs of the device. **b**, Responsivity as a function of V_{tg} for different V_{bg} measured at given f and T .

Reviewer #3:

Review of Nature Communications MS “Resonant Terahertz Detection Using Graphene Plasmons” by Bandurin et al.

This work presents an interesting study of resonant and non-resonant THz detection based on plasmons in bilayer graphene (BLG) – hBN based FET structure. The high quality BLG with mobility up to $10 \text{ m}^2/\text{Vs}$ (at $T = 10 \text{ K}$) at the doping density of $n = 10^{12} \text{ cm}^{-2}$ was used for the manufactured devices. At sub-THz frequencies (130 GHz), corresponding to the field oscillation periods longer than the plasmon damping time, the detection showed a well-known non-resonant overdamped behavior corresponding to the Q factor less than 1 in the FET channel. However, as the incident field frequency was increased to 2 THz, the clear resonant behavior of the detector was demonstrated, with the reasonably high Q factor between 4 and 11, ensuring significant resonant plasmon confinement in the FET channel.

I find this work interesting and timely. As the authors write, such an arrangement indeed allows the study of plasmon physics in confined geometries and under non-ambient conditions (e.g. low temperatures and high B-fields), without the need of tip-based spectroscopies.

We thank the Reviewer for careful reading of our manuscript and recognizing the important application of our detectors as a tool for fundamental plasmonics studies. With that respect, we would like to mention, that after the initial submission, we have carried out THz experiments in BLG/hBN superlattice devices and using our approach unveiled a new type of collective modes in moiré minibands as we now discuss in the revised manuscript in detail (Section: Miniband plasmons in graphene/hBN superlattices). We believe that these fundamentals observations are of particular importance for the part of plasmonics community dealing with graphene.

The paper is well-organized and is easy to read.

We thank the reviewer for this kind assessment.

I however question the application motivation for such devices, which only show the novel resonant behaviour at cryogenic temperatures. Further, when the authors mention the state of the art, they avoid the direct comparison between their devices and the state of the art in the literature.

We agree with the reviewer that a more detailed comparison of the state-of-the-art detector characteristics is required. As we were mostly focused on the plasmon resonances in graphene FETs we did not pay enough attention to this inquiry. Meanwhile, over the past month we have managed to improve the performance of our detectors drastically. This has been achieved in dual-gated BLG field effect transistors. We remind, that BLG is characterized by the gate-tuneable band structure such that one can open a band gap in its energy spectrum by applying a perpendicular electric field. This, in turn, substantially affects transistor nonlinearity characterized by the FET-factor introduced in the main text. Taking advantage of this observation, we equipped our new devices with local back gate and found a dramatic increase in overall responsivity in the broadband regime that exceeded 3 kV/W with the NEP of $0.2 \text{ pW/Hz}^{0.5}$ for certain combination of back and top gate voltages (Fig. S2b in the revised manuscript). We have revised our manuscript and included the results on

dual-gated photodetectors in the Supplementary Section 2. The achieved values of the responsivity and NEP are comparable to the commercial state-of-the-art cryogenic THz bolometers operating at the same f and T . We acknowledge this fact in the revised Supplementary Section 2.

I strongly recommend to update the manuscript with the table comparing such key parameters of the plasmonic detectors as NEP and the plasmon confinement λ_0 / λ_0 (for the corresponding operation temperature) for their devices and for the literature state of the art.

We thank the reviewer for this suggestion. In the revised Supplementary Section 2, we provide a table comparing NEP of our dual-gated devices with superconductor and semiconductor hot electron bolometers operating at the same f and T .

As per confinement ratio, we refrain from including this information into the table for two reasons. First of all, the most sensitive cryogenic detectors on the market are not plasmonic and therefore such comparison is irrelevant. Second, compression ratio is a property of a two-dimensional electronic system rather than a detector and will be the similar for plasmonic graphene devices of various types such as those employed in near-field studies (see e.g. M. Lundeberg et. al., Nat. Mater. **16**, 204–207 (2017)). We note, that in our work we provided these values to highlight the possibility to reach higher order plasmon modes and lower plasmon phase velocity and therefore ensure a strong confinement of electromagnetic fields, as until now it has not been yet reported for the case of BLG.

Additional comments:

The authors write that in monolayer graphene the electron mass is dependent on the electron density. This is rather confusing, since in the monolayer graphene, within the Dirac cone (i.e. in the range of about +/- 1.5 eV with respect to the neutrality point) the electron mass is zero and the band velocity is constant. I guess this statement should be revised or clarified.

Equation (2) of the main text, written in the form of $s = \sqrt{eV_g/m}$, contains the effective mass of charge carriers which, for the case of monolayer graphene (MLG), has to be replaced with the density-dependent cyclotron mass $m = p_F^2/v_F$. Here $v_F = 10^6$ m/s is the Fermi velocity, $p = \hbar\sqrt{\pi n}$ is the Fermi momentum. The use of this quantity is justified by the Drude formula for graphene conductivity (see Das Sarma et.al. Rev. Mod. Phys. **83** p. 407 (2011))

$$\sigma_{gr} = \frac{e^2}{2} \int_0^{+\infty} d\varepsilon D(\varepsilon) \frac{\mathbf{v}_p^2}{-i\omega + \tau_p^{-1}} \left(-\frac{\partial f_0}{\partial \varepsilon} \right) = \frac{ne^2}{\varepsilon_F / v_F^2} \frac{1}{-i\omega + \tau_{p_F}^{-1}}, \quad (\text{R1})$$

where $D(\varepsilon)$ is the density of states, $\mathbf{v}_p = v_F \mathbf{p} / p$ is the velocity vector for massless electrons, f_0 is the equilibrium electron distribution function. The comparison of Drude formula for

MLG and that for massive 2DEGs $\sigma_{2deg} = \frac{ne^2 / m}{-i\omega + \tau_{p_F}^{-1}}$, allows us to interpret the quantity ε_F / v_F^2

as an electron mass of its charge carriers. Note, the original derivation of (R1) assumed massless carriers stemming from familiar Dirac cones. Since the dispersion relation of plasmons depends only on electrodynamic properties of conductive layer (i.e., its

conductivity σ), the latter allows us to safely use the concept of mass in the form of $m = p_F/v_F$ when dealing with MLG.

Next, since $n \sim V_g$ and therefore $m \sim (V_g)^{1/2}$, one obtains that $s \sim (V_g)^{1/4}$ for gated MLG FETs. As per the bilayer graphene (BLG) case, $m_{BLG} = 0.036 m_e$ is density-independent which ensures faster $s \sim (V_g)^{1/2}$ dependence as pointed in our manuscript. We took advantage of this observation to vary the plasmon velocity over the wider range (compared to MLG) for fixed gate voltage span and thereby observe more resonances than anticipated for the case of MLG.

Further, what peak temperatures do the electron reach, and what is the ratio between the peak temperature and the Fermi temperature, in the presented THz detectors? It is well known that the transient electron heating in the THz fields can strongly modulate the conductivity in graphene (see e.g. Nature 561, 507 (2018), Nature Commun. 6, 7655 (2015)). Intuitively, this would positively add to the resistive self-mixing contribution in the FET, further enhancing the detector efficiency.

We agree with the Reviewer and appreciate this highly professional comment. Indeed, high frequency fields may cause significant heating of graphene's electrons especially at cryogenic temperatures. In the case of our antenna-coupled FETs, this heating stems from ac capacitive currents flowing between the source and gate terminals. The increase of electronic temperature may lead to a strong modification of graphene's conductivity and thereby affect the responsivity.

For the sake of demonstration, Fig. R4 shows $R_a(V_g)$ measured at different P of 129 GHz radiation. In agreement with the Reviewer's expectations, the responsivity magnitude changes drastically with increasing P . This happens precisely because BLG's conductivity is sensitive to the change in electronic temperature. To support this statement, we plot the FET factor obtained by measuring the sample's conductivity at different T . Clearly, the responsivity acquired at different P follows the evolution of the FET factor at various T . This is reflected in the shift and decrease of the responsivity magnitude with increasing P . We also refer to Fig. 2a of the main text which shows $R_a(V_g)$ at different T that resembles the behaviour found with respect to increase in the radiation power. Moreover, by comparing Fig. R4 and Fig. 2a of the main text one can estimate the peak electron temperature which for the case of the highest radiation power may reach that of liquid nitrogen (6 meV). As per the Fermi energy, it scales with carrier density and for a typical $n=10^{12} \text{ cm}^{-2}$ is of the order of 30 meV.

In our THz experiments we routinely take a great care of this effect by measuring the photovoltage ΔU at varying power P of incident radiation, as shown in Fig. R4, and limit ourselves to the P -range where ΔU scales linearly with power at all V_g thereby mitigating the change of graphene's conductivity. In the revised manuscript, we point to this fact (Section Methods).

Fig. R4. Role of electron heating. Normalized to unity responsivity as a function gate voltage measured in one of our BLG detectors at given T and f for different P . $P_0=4 \mu\text{W}$. Inset: FET-factor as a function of V_g for different T .

Quite a strong plasmonic confinement in the channel might indeed lead to a significant electron heating even at moderate powers of the incident THz signal. Has this effect been considered? Is it of relevance, or the relative temperature increase is small and can be neglected? A comment on this would be helpful.

We agree with the Reviewer, that a strong plasmonic confinement may lead to an increase of T_e as we discussed above and therefore affect the photoresponse. Nevertheless, as we have just mentioned, we report the data acquired at low enough P which does not change T_e drastically.

Typos:

- 1) Photovoltage-based spectroscopy of 2D plasmons
- 2) ... proportional to the the sensitivity
- 3) Reference list should be checked for accuracy (spelling of authors' names etc)

We are grateful for the Reviewer for pointing us these typos. We have amended the manuscript and corrected these misspellings.

REVIEWERS' COMMENTS:

Reviewer #2 (Remarks to the Author):

The authors have provided satisfactory replies to the reviewers' comments, and have made acceptable changes to the manuscript. The revised manuscript is expanded with further experimental results on structures with additional backgates which improve the detector responsivity. Additional results on moiré plasmons have been provided. I do not see any further objections to publication in Nature Communications.

Reviewer #3 (Remarks to the Author):

I agree with most of the responses by the Authors of this manuscript. However, I believe that the following points must be made more clear:

- 1) In the literature the term "Drude weight" is routinely used in conjunction with the eq. R1, without an introduction of an artificial (cyclotron) mass. I would suggest that the authors either use the Drude weight, or indeed introduce and explain the electron density-dependent "effective mass" (cyclotron mass) in MLG, $m = p_F/v_F$, in the same fashion as they did in their response to the Reviewers. I consider this as absolutely necessary, in order to avoid confusion among the broad readership of Nature Communications used to graphene featuring "massless Dirac fermions".
- 2) The table comparing the performance of graphene detectors from this work, and other existing technologies, must be placed in the main text, and not in a supplementary material. This comparison belongs to the main claims of this work, and therefore must be made directly visible to the readers.
- 3) I do not find the response of the Authors regarding the electron temperature increase compelling. The effect of the electron heating is strong, as evidenced in Fig. R4. The demonstrated detectors, when used by others, will not only operate in the regime of negligible electron heating, but also in the highly nonlinear regime as shown in Fig. R4. Therefore, the discussion of electron heating effect on the detector performance, and the results presented in Fig. R4, must be made known to the readers of this work, and therefore, must be included in the manuscript.

REVIEWERS' COMMENTS:

Reviewer #2 (Remarks to the Author):

The authors have provided satisfactory replies to the reviewers' comments, and have made acceptable changes to the manuscript. The revised manuscript is expanded with further experimental results on structures with additional backgates which improve the detector responsivity. Additional results on moiré plasmons have been provided. I do not see any further objections to publication in Nature Communications.

We thank the Reviewer for his/her kind support.

Reviewer #3 (Remarks to the Author):

I agree with most of the responses by the Authors of this manuscript. However, I believe that the following points must be made more clear:

1) In the literature the term "Drude weight" is routinely used in conjunction with the eq. R1, without an introduction of an artificial (cyclotron) mass. I would suggest that the authors either use the Drude weight, or indeed introduce and explain the electron density-dependent "effective mass" (cyclotron mass) in MLG, $m = p_F/v_F$, in the same fashion as they did in their response to the Reviewers. I consider this as absolutely necessary, in order to avoid confusion among the broad readership of Nature Communications used to graphene featuring "massless Dirac fermions".

We thank the Reviewer for this suggestion and provide a detailed explanation on why the eq. (2) of the main text is valid for monolayer graphene when the effective mass is replaced with the cyclotron mass. This is done in the revised main text and amended Supplementary Section 5.

2) The table comparing the performance of graphene detectors from this work, and other existing technologies, must be placed in the main text, and not in a supplementary material. This comparison belongs to the main claims of this work, and therefore must be made directly visible to the readers.

We thank the Reviewer for this suggestion and after very long negotiations between all the authors we came to the conclusion that such a table, if presented in the main text, would distract our readers from reaching quickly to the two main results of our work – resonant operation and miniband plasmons. We believe that, for the present work, the emphasis must be provided specifically to the resonances but not to the responsivity magnitude, and therefore retain only mentioning that the detectors' performance is exceptional and comparable to those available on the market. Note, the responsivity and determined NEP are explicitly reported in the main text and not hidden in the Supplementary Information. We believe, that a peculiar reader dealing with high-responsivity THz detection will find it easy to refer the table in the Supplementary Information.

3) I do not find the response of the Authors regarding the electron temperature increase compelling. The effect of the electron heating is strong, as evidenced in Fig. R4. The demonstrated detectors, when used by others, will not only operate in the regime of negligible electron heating, but also in the highly nonlinear regime as shown in Fig. R4. Therefore, the discussion of electron heating effect on the detector performance, and the results presented in Fig. R4, must be made known to the readers of this work, and therefore, must be included in the manuscript.

We agree with the Reviewer, and in the revised manuscript and added Supplementary Section 10, we state that the responsivity may vary upon increasing radiation power and include a detailed study of the detector performance outside the linear-in-P regime, similarly to what was done in Fig. R4. We are grateful for this suggestion.